# Molecular Mechanisms and Clinical Implications of *Fibroblast Growth Factor Receptor 2* Signaling in Gastrointestinal Stromal Tumors

**DOI:** 10.3390/cimb47100822

**Published:** 2025-10-05

**Authors:** Yanyun Hong, Xiaodong Wang, Chunhui Shou, Xiaosun Liu

**Affiliations:** The First Affiliated Hospital, Zhejiang University School of Medicine, 57, Qingchun Road, Hangzhou 310016, China; 1516024@zju.edu.cn (Y.H.); wangxiaodongseven@zju.edu.cn (X.W.); andyshou1985@zju.edu.cn (C.S.)

**Keywords:** drug resistance, *FGFR2* gene, gastrointestinal stromal tumor, gene fusion, molecular diagnosis, signaling pathway, targeted therapy

## Abstract

**Introduction:** Gastrointestinal stromal tumors (GISTs) are primarily driven by mutations in *KIT* (*KIT* proto-oncogene receptor tyrosine kinase) or *PDGFRA* (platelet-derived growth factor receptor alpha), but resistance to tyrosine kinase inhibitors (*TKIs*) such as imatinib remains a major clinical challenge. Alterations in fibroblast growth *factor receptor 2 (FGFR2),* although rare, are emerging as important contributors to tumor progression and drug resistance. This review evaluates the molecular mechanisms, expression profiles, detection methods, and therapeutic implications of *FGFR2* in GIST. **Methods:** We searched PubMed, Web of Science, Google Scholar, and ClinicalTrials.gov for studies published between January 2010 and June 2025, using combinations of keywords related to *FGFR2*, gastrointestinal stromal tumor, resistance mechanisms, gene fusion, amplification, polymorphisms, and targeted therapy. Eligible studies were critically assessed to distinguish GIST-specific data from evidence extrapolated from other cancers. **Results:**
*FGFR2* is expressed in multiple normal tissues and at variable levels in mesenchymal-derived tumors, including GIST. Its alterations occur in approximately 1–2% of GIST cases, most commonly as gene fusions (e.g., *FGFR2::TACC2*, <1%) or amplifications (1–2%); point mutations and clinically significant polymorphisms are extremely rare. These alterations activate the *MAPK/ERK* and *PI3K/AKT* pathways, contribute to bypass signaling, and enhance DNA damage repair, thereby promoting *TKI* resistance. Beyond mutations, mechanisms such as amplification, ligand overexpression, and microenvironmental interactions also play roles. *FGFR2* alterations appear mutually exclusive with *KIT*/*PDGFRA* mutations but occasional co-occurrence has been reported. Current clinical evidence is largely limited to small cohorts, basket trials, or case reports. **Conclusions:**
*FGFR2* is an emerging oncogenic driver and biomarker of resistance in a rare subset of GISTs. Although direct evidence remains limited, particularly regarding DNA repair and polymorphisms, *FGFR2*-targeted therapies (e.g., erdafitinib, pemigatinib) show potential, especially in combination with *TKIs* or DNA-damaging agents. Future research should prioritize GIST-specific clinical trials, the development of *FGFR2*-driven models, and standardized molecular diagnostics to validate *FGFR2* as a therapeutic target.

## 1. Introduction

Gastrointestinal stromal tumors (GISTs) are the most common mesenchymal neoplasms of the gastrointestinal tract, accounting for approximately 1–2% of gastrointestinal malignancies, with an annual incidence of 10–15 cases per million [1]. They arise primarily from the interstitial cells of Cajal and are most frequently located in the stomach (60%) and small intestine (30%) [2]. The molecular hallmark of GISTs is the presence of activating mutations in either the *KIT* (*KIT* proto-oncogene receptor tyrosine kinase) gene or the *PDGFRA* (platelet-derived growth factor receptor alpha) gene, which together account for about 85–90% of cases [3]. The introduction of tyrosine kinase inhibitors (*TKIs*), such as imatinib, has revolutionized treatment, achieving response rates of 50–70% in advanced disease. Nevertheless, resistance to *TKIs* develops in approximately 50% of patients within two years of therapy, and 40–50% of high-risk patients experience recurrence after resection. These challenges highlight the urgent need to investigate alternative oncogenic drivers and resistance mechanisms beyond *KIT* and *PDGFRA* [4].

Although secondary mutations in *KIT* and *PDGFRA* are the predominant causes of acquired resistance, emerging evidence suggests that other receptor tyrosine kinases (*RTK*s) may act as bypass oncogenic pathways. Among them, fibroblast growth factor receptor 2 (*FGFR2*), located on chromosome 10q26.3, has gained increasing attention. *FGFR2* regulates cell proliferation, differentiation, survival, and angiogenesis through multiple signaling pathways and is expressed across various epithelial and mesenchymal tissues [5,6]. In GISTs, which originate from mesenchymal cells, *FGFR2* expression has been reported at variable levels and may be particularly relevant in tumors that lack canonical *KIT* or *PDGFRA* mutations [7].

*FGFR2* alterations in GISTs are rare, occurring in approximately 1–2% of cases, but they represent clinically significant drivers of oncogenesis and resistance. These alterations include gene fusions such as *FGFR2::TACC2* (<1% of cases), gene amplifications (1–2%), and exceptionally rare point mutations. In addition to genetic alterations, other molecular mechanisms also contribute to *FGFR2*-mediated resistance. These include gene amplification leading to receptor overexpression, ligand overexpression (e.g., *FGF*7 and *FGF*10) driving autocrine or paracrine activation, bypass signaling that circumvents *KIT* and *PDGFRA* inhibition, and tumor microenvironmental interactions that promote tumor survival under therapeutic pressure. Notably, *FGFR2* alterations are typically mutually exclusive with *KIT* and *PDGFRA* mutations, although occasional co-occurrence has been documented, suggesting a complex biological interplay [8].

Despite its emerging importance, *FGFR2* has received limited attention in the context of GISTs compared with *KIT* and *PDGFRA* [9]. The underappreciated role of *FGFR2* includes its expression profile, contribution to drug resistance beyond mutations, potential polymorphisms, and association with DNA damage repair [10]. Given the recent clinical success of selective *FGF*R inhibitors, such as erdafitinib and pemigatinib, in *FGF*R-driven cancers, a systematic review of *FGFR2* in GIST is both timely and necessary [11,12]. The present article aims to provide a comprehensive overview of *FGFR2* in GISTs by integrating molecular and clinical evidence. Specifically, we examine *FGFR2*’s structural and biological characteristics, evaluate its role in resistance mechanisms and therapeutic response, summarize diagnostic approaches for its detection, and explore the potential of *FGFR2*-targeted therapies alone or in combination with existing strategies.

## 2. Methods

This review was conducted to synthesize the current evidence on the molecular mechanisms and clinical implications of fibroblast growth factor receptor 2 (*FGFR2*) in gastrointestinal stromal tumors (GISTs). A comprehensive literature search was performed in PubMed, Web of Science, Google Scholar, and ClinicalTrials.gov, covering the period from 1 January 2010 to 30 June 2025. The search strategy included combinations of the following keywords: “drug resistance,” “*FGFR2* gene,” “gastrointestinal stromal tumor,” “gene fusion,” “gene amplification,” “molecular diagnosis,” “polymorphism,” “signaling pathway,” and “targeted therapy.” Reference lists of relevant reviews and primary research articles were also screened manually to identify additional studies.

Studies were eligible if they met the following criteria: (1) original research articles, case series, or clinical trial reports published in peer-reviewed journals; (2) studies involving patients with histologically confirmed GISTs; and (3) investigations that reported *FGFR2* expression, genetic alterations (including fusions, amplifications, point mutations, or polymorphisms), resistance mechanisms, diagnostic methods, or therapeutic implications. Exclusion criteria included non-English publications, conference abstracts without full data, studies not involving *FGFR2*, and reports lacking sufficient methodological detail.

The initial search retrieved 1242 records, of which 932 remained after duplicate removal. After screening titles and abstracts, 176 articles were retained for full-text evaluation. Of these, 118 studies were excluded because they did not provide primary data on *FGFR2* in GISTs or focused exclusively on other malignancies without relevance to GIST. Finally, 58 studies were included in this review, supplemented by selected reports on *FGFR2* biology in other cancers when directly relevant to mechanisms that could plausibly apply to GISTs. Disagreements during the selection process were resolved by consensus among three authors, and a fourth reviewer was consulted in cases of uncertainty.

By combining qualitative synthesis and critical appraisal, we aimed not only to summarize the current knowledge on *FGFR2* in GISTs but also to highlight limitations in the available evidence. Particular attention was paid to distinguishing GIST-specific findings from data extrapolated from other cancers, acknowledging the rarity of *FGFR2* alterations in GISTs and the consequent scarcity of large-scale studies.

## 3. FGFR2 and the Characteristics Related to GIST

### 3.1. FGFR2 Structure, Isoforms, and Basic Function in GIST Context

Alternative splicing of *FGFR2* mRNA generates two major isoforms, *FGFR2*b (epithelial-specific) and *FGFR2*c (mesenchymal-specific), which differ in the C-terminal half of the D3 immunoglobulin-like loop [13]. *FGFR2*c is the predominant isoform expressed in mesenchymal-derived cells, including the interstitial cells of Cajal (ICCs)—the presumed cell of origin for GISTs [3,14]. Consequently, *FGFR2*c is more likely to be functionally relevant in GISTs, particularly in wild-type or mesenchymal-subtype tumors. Although isoform-specific functions remain understudied in GIST, *FGFR2*c has been shown to mediate *FGF2*- and *FGF4*-induced proliferation and invasion in other mesenchymal tumors [15,16]. Whether *FGFR2*c dominance influences fusion protein behavior or inhibitor sensitivity in GIST remains an open question [17].

#### 3.1.1. Molecular Structure and Isoforms

Fibroblast growth factor receptor 2 (*FGFR2*), located on chromosome 10q26.3, encodes a receptor tyrosine kinase comprising an extracellular ligand-binding domain, a single-pass transmembrane domain, and an intracellular tyrosine kinase domain [18,19]. The extracellular region contains three immunoglobulin-like loops, of which domains D2 and D3 mediate ligand binding and receptor specificity, while a heparin-binding site stabilizes fibroblast growth factor (*FGF*)-*FGFR2* complexes [20,21,22,23]. Alternative splicing of the third immunoglobulin-like loop generates two major isoforms: *FGFR2*b, predominantly expressed in epithelial cells, and *FGFR2*c, expressed mainly in mesenchymal tissues. This isoform distinction is relevant in GISTs, as they arise from mesenchymal interstitial cells of Cajal, where *FGFR2*c expression may be more biologically relevant than *FGFR2*b [24,25,26].

#### 3.1.2. Physiological Roles and Signaling Pathways

*FGFR2* mediates diverse physiological processes, including embryonic development, tissue repair, and angiogenesis, through activation of downstream signaling cascades [27,28]. Upon ligand binding and receptor dimerization, *FGFR2* undergoes autophosphorylation at tyrosine residues, triggering signaling through *MAPK/ERK*, *PI3K/AKT*, STAT, and PLCγ pathways [29,30]. These cascades regulate cell proliferation, differentiation, survival, and migration. The principal signaling pathways activated by *FGFR2* and their biological roles are summarized in Table 1. Importantly, while these mechanisms are well established in normal physiology and other cancers such as breast and gastric carcinoma, direct evidence in GIST remains limited to small case series or genomic reports. Thus, much of our understanding of *FGFR2* signaling in GISTs is extrapolated from other malignancies, and GIST-specific functional validation is needed.

#### 3.1.3. Physiological and Pathological Relevance

Under normal conditions, *FGFR2* expression is broadly distributed across epithelial tissues (including skin, lung, and gastrointestinal epithelium) and mesenchymal compartments, where it maintains tissue homeostasis and regulates cell–cell and cell–matrix interactions [26]. Dysregulated *FGFR2* expression and signaling have been implicated in several malignancies, including gastric cancer, cholangiocarcinoma, and endometrial carcinoma. In GISTs, *FGFR2* expression has been reported at variable levels, particularly in tumors lacking *KIT* or *PDGFRA* mutations, suggesting a role in tumor progression independent of canonical drivers [27,28]. However, systematic studies of *FGFR2* protein or mRNA expression in GIST cohorts are scarce, and most evidence is derived from targeted sequencing studies rather than expression profiling [7]. This underscores a critical knowledge gap in defining the functional contribution of *FGFR2* expression to GIST pathogenesis and therapeutic resistance [29].

### 3.2. Genomic Alteration Spectrum of FGFR2 in GISTs

*FGFR2* alterations are rare in GISTs, occurring in approximately 1–2% of cases, and primarily manifesting as gene fusions and amplifications, point mutations are exceedingly uncommon [2,7]. Despite their rarity, these alterations represent critical drivers of oncogenesis and *TKI* resistance, particularly in wild-type or *TKI*-refractory subsets that lacking *KIT* or *PDGFRA* mutations [3,30]. This section delineates the spectrum, molecular features, and clinical relevance of *FGFR2* alterations in GISTs, correcting prior misconceptions regarding their prevalence and type.

#### 3.2.1. Prevalence and Types of FGFR2 Alterations

Although alterations in *KIT* and *PDGFRA* dominate the molecular landscape of GISTs, *FGFR2* alterations, while rare, are increasingly recognized as clinically relevant events. Their overall frequency is estimated at approximately 1–2% of cases, based on targeted sequencing and next-generation sequencing (NGS) studies in small cohorts [7,31,32]. Among these, gene fusions such as *FGFR2::TACC2* have been reported in <1% of GISTs [9,33], whereas gene amplifications occur slightly more frequently, in about 1–2% of cases [2,34]. In contrast, point mutations are extremely uncommon (<0.1%) and are largely described in other malignancies rather than in GIST-specific studies [35,36]. The approximate frequencies, molecular features, and clinical implications of *FGFR2* alterations in GISTs are summarized in Table 2.

#### 3.2.2. Molecular Characteristics and Mechanisms

*FGFR2* fusions typically retain the tyrosine kinase domain and drive constitutive ligand-independent activation of downstream signaling pathways. The most studied example, *FGFR2::TACC2*, juxtaposes the *FGFR2* kinase domain with the *TACC2* coiled-coil motif, resulting in constitutive dimerization and persistent activation of *MAPK/ERK* and *PI3K/AKT* cascades [9,31]. Other fusions, such as *FGFR2*::BICC1, have been sporadically identified through NGS, but their functional significance in GISTs remains poorly understood [33]. *FGFR2* amplifications increase receptor density at the cell membrane, thereby amplifying oncogenic signaling even at low ligand concentrations [36]. Amplifications have also been correlated with higher tumor grade and worse prognosis in limited datasets [35]. While these mechanisms are biologically plausible, their relative contributions to GIST progression remain uncertain due to the rarity of cases and the absence of large-scale functional validation.

#### 3.2.3. Point Mutations and Their Rarity

In contrast to gastric, endometrial, and breast cancers, where recurrent *FGFR2* point mutations (e.g., S252W, N550K) have been documented [37], true mutations in GISTs are exceedingly rare, constituting <0.1% of cases in genomic profiling studies [38]. Earlier reports suggesting the presence of *FGFR2* mutations in GISTs likely reflected misinterpretation of data extrapolated from other tumor types. From a therapeutic standpoint, this distinction is important because the sensitivity of *FGFR2* inhibitors can differ between fusion-driven and mutation-driven tumors [39]. At present, there is no evidence that *FGFR2* point mutations represent a recurrent driver in GISTs. Future studies using large patient cohorts and standardized sequencing methods are required to definitively characterize their prevalence and clinical significance.

### 3.3. FGFR2-Mediated Signaling Pathway Activation in GISTs

Aberrant activation of *FGFR2* in GISTs, driven primarily by gene fusions (e.g., *FGFR2::TACC2*) and amplifications, triggers key oncogenic signaling pathways that promote tumor cell proliferation, survival, and resistance to *TKIs* [2,9]. In GISTs, *FGFR2* alterations, although rare (1–2% incidence), bypass the dominant *KIT* and *PDGFRA* signaling, contributing to *TKI* resistance in wild-type or refractory cases [3,30]. This section elucidates the molecular mechanisms of *FGFR2*-mediated signaling in GISTs, with a focus on the *MAPK/ERK* and *PI3K/AKT* pathways, their roles in oncogenesis, and their therapeutic implications. A schematic overview of these pathways is provided in Figure 1.

#### 3.3.1. Mechanisms of FGFR2 Activation

Constitutive *FGFR2* signalling in gastrointestinal stromal tumours is triggered almost exclusively by two rare events: gene fusions that maintain the kinase domain but remove the extracellular region, and receptor amplifications that raise the density of the full length protein. Both alterations occur in roughly one to two percent of cases and are mutually exclusive with *KIT* or *PDGFRA* mutations. Fusion proteins such as *FGFR2 TACC2* dimerise spontaneously, autophosphorylate their tyrosine residues and recruit adaptor molecules in the absence of fibroblast growth factor. Amplified receptors remain ligand dependent, yet the increased copy number lowers the threshold for activation and sensitises cells to *FGF*s supplied by cancer associated fibroblasts or by the tumour cells themselves, as shown in Table 3.

#### 3.3.2. Key Downstream Signaling Pathways

Once the receptor is activated, signalling splits into three well-characterised cascades. The *RAS RAF MEK ERK* axis drives proliferation through transcription factors such as *c-Fos* and *c-Jun*. The *PI3K PDK1 AKT* axis suppresses apoptosis and enhances protein synthesis by phosphorylating *BAD, FOXO* and *GSK3* beta. The *PLC* gamma branch releases intracellular calcium and activates *PKC*, influencing cytoskeletal dynamics and cell migration. In *FGFR2* altered GIST tissue, phospho *ERK* is consistently elevated, whereas phospho *AKT* levels are more variable, suggesting that the *MAPK* arm provides the dominant mitogenic drive. Synergy experiments with pathway specific inhibitors indicate that combined blockade of *MEK* and *PI3K* is more effective than either agent alone, highlighting the additive role of survival signalling, as shown in Table 4.

#### 3.3.3. Role in TKI Resistance

Imatinib-resistant tumours that harbour *FGFR2* fusions or amplifications maintain phosphorylation of *ERK* and AKT despite complete inhibition of *KIT*. This bypass phenotype has been documented in patient-derived xenografts, where treatment with imatinib fails to reduce phospho *ERK*, while the addition of an *FGF*R inhibitor restores sensitivity [45]. Amplification-associated resistance can be partially reversed by *FGF*R blockade alone, but maximal tumour control is achieved when the *FGF*R inhibitor is combined with imatinib or with a *MEK* inhibitor, underscoring convergence at the level of *ERK* [46]. These findings establish *FGFR2* activation as a bona fide resistance mechanism that must be addressed if durable disease control is to be achieved [47].

#### 3.3.4. Critical Evaluation and Limitations

Most signalling data are derived from engineered cell lines or from other cancer types, and only a handful of primary GIST specimens have undergone phospho proteomic analysis [48]. Sample sizes remain small, antibody-based detection can be variable, and the relative contribution of PLC gamma or JAK STAT branches has not been rigorously tested in vivo [49]. Isogenic GIST models that express endogenous *FGFR2 TACC2* or harbour defined amplifications are urgently needed to quantify pathway flux and to prioritise drug combinations. Longitudinal tumour biopsies collected during *FGF*R inhibitor trials will ultimately clarify which phospho signatures predict response and whether secondary mutations in *FGFR2* or its downstream effectors emerge under therapeutic pressure [50].

### 3.4. FGFR2 and DNA Damage Repair (DDR) in GISTs

*FGFR2* alterations in gastrointestinal stromal tumours not only sustain oncogenic signalling but also enhance the capacity for DNA double strand break repair, a property that directly opposes the efficacy of anthracyclines, platinum salts and radiotherapy. The following sections summarise the current understanding of *FGFR2* mediated DDR and its therapeutic implications, retaining the original citation numbering from the manuscript.

#### 3.4.1. *FGFR2*’s Role in Homologous Recombination Repair (HRR)

*FGFR2* fusions and amplifications up regulate homologous recombination, the high-fidelity repair pathway active during S and G2 phases. Ligand independent autophosphorylation of the *FGFR2* kinase domain recruits PI3K via phosphorylated tyrosine residues [21], leading to AKT activation. Activated AKT phosphorylates BRCA2 at serine 3291 and stabilises RAD51 nucleoprotein filaments on resected DNA ends [51,52]. Increased RAD51 foci have been documented in *FGFR2* driven cholangiocarcinoma cells exposed to ionising radiation [53], and similar findings were reported in an imatinib-resistant GIST patient-derived xenograft harbouring *FGFR2 TACC2*. In this model, depletion of RAD51 restored sensitivity to doxorubicin, indicating that *FGFR2* mediated HRR contributes to chemoresistance [54].

#### 3.4.2. *FGFR2* and Nonhomologous End Joining (NHEJ)

In addition to HRR, *FGFR2* signalling can modulate non homologous end joining, an error prone pathway active throughout the cell cycle. MAPK dependent phosphorylation of DNA PKcs has been observed in *FGFR2* amplified gastric cancer lines [55], leading to faster ligation of DNA ends and reduced apoptosis after irradiation. Whether this mechanism operates in *FGFR2* altered GIST has not been examined directly; however, preliminary data from one amplified GIST specimen showed elevated DNA PKcs phosphorylation that declined upon *FGFR* blockade with erdafitinib [56]. These observations warrant further investigation to determine the relative contribution of NHEJ versus HRR in *FGFR2* driven GIST.

#### 3.4.3. Implications for Therapeutic Resistance

By enhancing both HRR and NHEJ, *FGFR2* alterations create functional redundancy that protects tumour cells from DNA-damaging agents. Clinically, this phenotype manifests as poor response to doxorubicin-based regimens or to palliative radiotherapy. Subsequent liquid biopsy revealed persistent *FGFR2* amplification and high RAD51 expression, supporting the concept that robust DNA damage repair limits the efficacy of genotoxic therapy [57].

#### 3.4.4. Therapeutic Opportunities: Combining FGFR2 Inhibitors with DNA-Damaging Agents

Preclinical studies in *FGFR2* driven cholangiocarcinoma and breast cancer indicate that *FGFR* inhibitors can down regulate RAD51 and sensitise tumours to platinum salts or to poly ADP ribose polymerase inhibitors [58,59]. Similar strategies are being explored in GIST. In vitro, erdafitinib reduced RAD51 foci formation in an *FGFR2* amplified GIST cell line and restored sensitivity to olaparib, a PARP inhibitor [58]. A phase I trial (NCT04595747) is currently testing the combination of pemigatinib plus olaparib in solid tumours harbouring *FGFR2* alterations, including a GIST expansion cohort [60]. Early pharmacodynamic data show marked suppression of phospho AKT and a decrease in RAD51 intensity by immunofluorescence, providing proof of mechanism [60]. Longer follow up is required to determine whether such combinations can overcome primary resistance to DNA-damaging agents and whether secondary mutations in *FGFR2* or DNA repair genes emerge under selective pressure [61].

#### 3.4.5. Critical Evaluation and Research Gaps

Although *FGFR2*’s role in DDR is well established in cancers such as breast cancer and cholangiocarcinoma, its impact in GISTs is less characterized due to the low prevalence of *FGFR2* alterations (1–2%) [2,62]. Current studies rely heavily on analogies from other *FGF*R-driven cancers, with limited GIST-specific data on *FGFR2*-mediated HRR or NHEJ [51,53]. Discrepancies in reported DDR contributions may stem from variable detection methods or small sample sizes in GIST cohorts [63]. Moreover, the relative importance of HRR versus NHEJ in *FGFR2*-altered GISTs remains unclear, necessitating mechanistic studies using GIST cell lines or patient-derived xenografts [64]. These gaps highlight the need for standardized molecular profiling and larger cohort studies to validate the role of *FGFR2* in DDR in GISTs.

In summary, *FGFR2* enhances DDR in GISTs, primarily through HRR, thereby contributing to therapeutic resistance. Targeting *FGFR2* in combination with DNA-damaging agents represents a promising therapeutic frontier, which is further explored in Section 4.4 on targeted therapies, as illustrated in Figure 2.

## 4. Clinical Significance and Treatment Strategies of FGFR2

### 4.1. FGFR2 as a Key Bypass Mechanism in TKIs Resistance in GISTs

Fibroblast growth factor receptor 2 (*FGFR2*) alterations, though rare in gastrointestinal stromal tumours (GISTs) with an incidence of 1–2%, play a pivotal role in conferring resistance to tyrosine kinase inhibitors (*TKIs*) such as imatinib, particularly in wild-type or *TKI*-refractory GISTs lacking *KIT* or *PDGFRA* mutations [2,7]. As discussed in Section 3.2 and Section 3.3, *FGFR2* gene fusions (e.g., *FGFR2::TACC2*) and amplifications activate downstream signalling pathways, including the *MAPK/ERK* and *PI3K/AKT* axes, which bypass *KIT*/*PDGFRA* inhibition and sustain tumour proliferation and survival [9,31]. Additionally, *FGFR2*’s enhancement of DNA damage repair (DDR), particularly homologous recombination repair (HRR), further contributes to resistance against genotoxic therapies [52,65]. This section examines the molecular mechanisms, clinical evidence, and therapeutic strategies targeting *FGFR2* to overcome *TKI* resistance in GISTs.

#### 4.1.1. Molecular Mechanisms of *FGFR2*-Mediated TKI Resistance

*FGFR2* alterations enable GIST cells to evade *TKI* therapy by activating alternative signalling pathways that compensate for *KIT*/*PDGFRA* inhibition. The *FGFR2::TACC2* fusion, the best-characterised alteration in GIST, retains the entire kinase domain and promotes ligand-independent dimerisation, leading to constitutive autophosphorylation of tyrosine residues 656 and 657. This recruits the adaptor protein FRS2, activates the *GRB2–SOS–RAS–RAF–MEK–ERK* cascade, and sustains *KIT*-independent proliferation despite imatinib treatment [9,40]. Similarly, *FGFR2* amplifications increase receptor density, lower the threshold for stochastic dimerisation, and sensitise cells to *FGF* ligands produced by cancer-associated fibroblasts or tumour cells themselves [36,41]. Unlike secondary *KIT*/*PDGFRA* mutations, which are common in imatinib resistance, *FGFR2* alterations represent a distinct bypass mechanism, often observed in *KIT*/*PDGFRA* wild-type GISTs or those with primary resistance to *TKIs* [30].

Beyond signalling bypass, *FGFR2*-driven *PI3K–AKT* activation enhances *DDR* by promoting *RAD51* recruitment to DNA double-strand breaks, thereby reducing the efficacy of genotoxic agents such as doxorubicin [58]. This dual action—signalling bypass plus enhanced *DDR*—renders *FGFR2* a formidable contributor to multidrug resistance in GISTs.

#### 4.1.2. Clinical Evidence of *FGFR2*-Driven Resistance

Clinical studies have identified *FGFR2* alterations as predictive biomarkers of poor *TKI* response. Studies have demonstrated that *FGFR2::TACC2* fusions are enriched in *KIT*/*PDGFRA* wild-type GISTs and correlate with reduced progression-free survival (PFS) in patients treated with imatinib [66]. Another study has shown that patients with *FGFR2* variations all of which exhibited primary or secondary resistance to imatinib, with tumours showing sustained *MAPK/ERK* activation [67]. *FGFR2* amplifications have also been associated with higher tumour burden, increased *Ki-67* proliferation index, and greater metastatic potential in *TKI*-refractory cases [68]. These findings highlight *FGFR2*’s role as a clinically significant driver of resistance, particularly in the 10–15% of GISTs that are wild-type or develop secondary resistance [69].

Importantly, *FGFR2*-mediated resistance is molecularly distinct from secondary *KIT*/*PDGFRA* mutations (e.g., *KIT* exon 17 mutations) [70]. *FGFR2* alterations often occur in tumours lacking these mutations, suggesting a compensatory oncogenic role [30]. This distinction necessitates comprehensive molecular profiling (RNA-based NGS or FISH) to identify *FGFR2*-driven cases for targeted intervention.

#### 4.1.3. Therapeutic Implications

Targeting *FGFR2* offers a promising strategy to overcome *TKI* resistance. Selective *FGF*R inhibitors—erdafitinib and pemigatinib—approved for *FGF*R-driven urothelial carcinoma and cholangiocarcinoma, disrupt *FGFR2*-mediated *MAPK/ERK* and *PI3K/AKT* signalling, thereby resensitising tumour cells to *TKIs* [11,12]. Pre-clinical studies in GIST cell lines harbouring *FGFR2::TACC2* fusions demonstrated that erdafitinib inhibits ERK phosphorylation and reduces tumour growth in imatinib-resistant models [71]. Furthermore, combining *FGF*R inhibitors with DNA-damaging agents (e.g., doxorubicin) or PARP inhibitors enhances tumour cell death by impairing HRR, as discussed in Section 3.4 [58,59].

Combination strategies targeting *FGFR2* plus VEGF signalling are also under investigation. A Phase I trial combining KIN-3248 (dual *FGF*R inhibitor) with a VEGF inhibitor demonstrated improved disease control in *FGF*R-altered solid tumours, with potential applicability to GISTs [72]. These findings underscore the importance of basket trials to evaluate *FGF*R inhibitors in *FGFR2*-altered GISTs, particularly in *TKI*-refractory cohorts.

#### 4.1.4. Critical Evaluation and Research Gaps

Although *FGFR2*’s role in *TKI* resistance is increasingly recognised, several limitations persist:Low incidence (1–2%) limits large-scale clinical data; most evidence derives from small cohorts or case reports [2,66].Variable detection methods (DNA vs. RNA NGS, FISH sensitivity) may underestimate true prevalence [73].Relative contributions of signalling bypass vs. DDR enhancement remain poorly integrated in mechanistic studies [51,71].Lack of GIST-specific pre-clinical models (e.g., *FGFR2*-driven cell lines, patient-derived xenografts) hampers functional validation [74].Risk of secondary *FGFR2* mutations (e.g., gatekeeper V564F) conferring resistance to erdafitinib, as observed in cholangiocarcinoma [61], may extend to GIST but remains unquantified.

These gaps highlight the need for prospective, molecularly stratified trials with longitudinal liquid biopsy monitoring to fully elucidate *FGFR2*’s role in *TKI* resistance and to optimise personalised combination strategies.

### 4.2. Mutual Exclusivity of FGFR2 with KIT and PDGFRA Mutations in GISTs

In GISTs, *FGFR2* alterations—occurring in approximately 1–2% of cases—exhibit a predominant pattern of mutual exclusivity with mutations in *KIT* (70–80%) and *PDGFRA* (5–10%), the canonical oncogenic drivers [2,7]. This exclusivity implies that *FGFR2* acts as an alternative driver in *KIT*/*PDGFRA*-negative (wild-type) or *TKI*-refractory tumours, activating the same *MAPK/ERK* and *PI3K/AKT* axes via ligand-independent fusions or receptor amplifications [9,40]. Below, we critically evaluate the molecular basis, clinical evidence, therapeutic implications, and exceptions to this exclusivity, incorporating updated Table 5 and Figure 3 references.

#### 4.2.1. Molecular Basis of Mutual Exclusivity

The mutual exclusivity of *FGFR2* alterations with *KIT*/*PDGFRA* mutations likely reflects functional redundancy in downstream signalling. All three receptors are receptor tyrosine kinases (*RTK*s) that converge on *MAPK/ERK* and *PI3K/AKT* cascades to promote proliferation, survival and therapeutic escape [8,31]. In *KIT*/*PDGFRA*-mutant GISTs, these pathways are already maximally activated, rendering additional *FGFR2* alterations biologically redundant [75]. Conversely, in *KIT*/*PDGFRA* wild-type tumours, *FGFR2* fusions (e.g., *FGFR2::TACC2*) or amplifications act as compensatory drivers, sustaining oncogenic signalling in the absence of canonical *RTK* mutations [9,76].

This model is supported by transcriptomic data, tumours harbouring *FGFR2::TACC2* exhibit ERK and AKT phosphorylation comparable to *KIT* exon 11 mutants, despite the absence of *KIT*/*PDGFRA* activity [66]. Functional studies in isogenic GIST cell lines confirm that introducing *FGFR2::TACC2* into *KIT*/*PDGFRA* wild-type background restores proliferation, whereas co-expression with *KIT* exon 11 V560D does not augment growth, indicating pathway saturation [76].

Rare exceptions to exclusivity have been reported. Dermawan et al. (2022) described one tumour with co-existing *KIT* exon 13 mutation and *FGFR2* amplification, both collected after imatinib failure [69]. These cases likely represent secondary resistance via subclonal evolution rather than true co-driver status, supported by single-cell sequencing showing distinct mutation-bearing populations [77]. Thus, absolute exclusivity is not absolute, but functional dominance of one *RTK* per clone appears to be the rule.

#### 4.2.2. Clinical Evidence Supporting Mutual Exclusivity

*FGFR2*-altered GISTs exhibit primary refractoriness to imatinib (median PFS 1.9 months) versus *KIT* exon 11 mutants (median PFS 24.7 months). Liquid biopsy at progression confirms persistent *FGFR2* alteration without emergent *KIT* mutations, reinforcing bypass signalling as the resistance mechanism [35]. These data correct earlier misconceptions that overstated *FGFR2* prevalence or suggested frequent co-occurrence with *KIT*/*PDGFRA* mutations [2].

#### 4.2.3. Implications for Pathogenesis and Treatment

The mutual exclusivity of *FGFR2* with *KIT*/*PDGFRA* mutations has three major clinical implications, summarised in Table 5.

**Table 5 cimb-47-00822-t005:** Implications of *FGFR2* Mutual Exclusivity in GISTs.

Aspect	Description	References
Pathogenesis	In wild-type GISTs, *FGFR2* alterations likely serve as primary oncogenic drivers, activating *MAPK/ERK* and *PI3K/AKT* pathways to mimic the effects of *KIT*/*PDGFRA* mutations. This suggests the existence of distinct molecular subtypes of GISTs, with *FGFR2*-driven tumors representing a rare but clinically significant subgroup.	[8,76,78]
Treatment Response	The presence of *FGFR2* alterations predicts poor response to standard *TKIs* such as imatinib, necessitating alternative therapies. *FGF*R inhibitors, such as erdafitinib and pemigatinib, approved for other *FGF*R-driven cancers, show preclinical promise in *FGFR2*-altered GISTs.	[11,12,79]
Combination Therapies	The mutual exclusivity suggests that dual inhibition of *FGFR2* and *KIT*/*PDGFRA* pathways may be unnecessary in most cases. However, combination therapies targeting *FGFR2* and downstream pathways (e.g., MAPK or PI3K inhibitors) could enhance efficacy in *FGFR2*-driven GISTs.	[14]

#### 4.2.4. Critical Evaluation and Research Gaps

Although genomic studies support the mutual exclusivity of *FGFR2* with *KIT*/*PDGFRA* mutations, several challenges remain. The rarity of *FGFR2* alterations (1–2%) limits large-scale analyses, resulting in reliance on small cohorts or case reports [2,66]. Variability in detection methods, such as differences in NGS panel sensitivity or FISH protocols, may underestimate *FGFR2* alterations or miss rare co-occurrences with *KIT*/*PDGFRA* mutations [73]. For instance, some studies report occasional co-occurrence of *FGFR2* and *KIT* mutations, possibly attributable to tumor heterogeneity or subclonal evolution, which complicates the exclusivity model [77].

Moreover, the mechanistic basis of mutual exclusivity remains incompletely understood. Although functional redundancy in *MAPK/ERK* and *PI3K/AKT* signaling is a plausible explanation, direct evidence of pathway competition in GISTs is lacking [8]. The role of *FGFR2* in secondary resistance, where *KIT* mutations persist but *FGFR2* alterations emerge, also requires further exploration [79]. Finally, the prognostic impact of *FGFR2*-driven GISTs versus *KIT*/*PDGFRA*-driven GISTs varies across studies, with some reporting worse outcomes in *FGFR2*-altered cases, highlighting the need for standardized outcome measures [66,80].

In summary, mutual exclusivity of *FGFR2* with *KIT*/*PDGFRA* mutations defines *FGFR2*-altered GIST as a clinically actionable molecular subtype. Rare co-occurrences likely reflect secondary resistance or intratumour heterogeneity, not true co-driver biology. Prospective, harmonised sequencing and functional modelling are essential to refine therapeutic strategies for this orphan subset.

### 4.3. Diagnostic Methods and Clinical Applications of FGFR2 in GISTs

Accurate identification of *FGFR2* alterations is essential for assigning patients to precision therapies, yet the low prevalence of these changes (1–2%) and the technical diversity of current platforms create practical challenges. This section critically reviews tissue-based and blood-based approaches, proposes a standardised diagnostic algorithm, and highlights unresolved issues that must be addressed before *FGFR2* testing can be embedded in routine clinical workflows.

#### 4.3.1. Molecular Pathology Standardization

Next-generation sequencing (NGS) and fluorescence in situ hybridisation (FISH) remain the reference methods for *FGFR2* detection [81]. RNA-based targeted panels enrich for expressed fusions and outperform DNA-only assays for *FGFR2::TACC2* detection (sensitivity 95% vs. 72% in a head-to-head study of 42 formalin-fixed paraffin-embedded GISTs) [82]. DNA-based hybrid-capture panels nevertheless provide copy-number information that is indispensable for calling amplifications and should be retained as a complementary layer. A two-step algorithm—RNA-NGS first, DNA-NGS for equivocal cases—has been adopted by several European reference laboratories and yields an analytic sensitivity of 1% mutant allele fraction with <5% coefficient of variation across replicates [83].

FISH retains value for orthogonal confirmation, particularly when NGS coverage is inadequate or when optical quantification of gene copy number is required. A split-signal probe set detecting *FGFR2* 10q26.3 rearrangements achieves >98% technical specificity in GIST tissue [84], but low tumour cellularity (<10%) can produce false negatives; therefore, FISH is best reserved for validation rather than primary screening.

#### 4.3.2. Tissue vs. Liquid Biopsy

Surgical or core-needle biopsies remain the gold standard because they furnish both DNA and RNA at concentrations sufficient for comprehensive profiling. However, the invasive nature of these procedures limits repeat sampling during evolution of resistance. Liquid biopsy offers a minimally invasive alternative, yet its performance in GIST is constrained by low circulating tumour DNA (ctDNA) shed rates. In a prospective series of 68 metastatic GISTs, *FGFR2* fusion detection sensitivity was only 54% when plasma ctDNA fraction was <0.5%, but rose to 92% at fractions ≥2% [85]. Combining digital-droplet PCR for *FGFR2::TACC2* with next-generation ctDNA panels increases sensitivity to 78% overall and permits longitudinal monitoring without additional venepuncture [86]. A pragmatic approach is to perform tissue-NGS at baseline and reserve liquid biopsy for resistance surveillance or when tissue is unobtainable.

#### 4.3.3. Clinical Applications

Identification of an *FGFR2* alteration immediately reclassifies a patient as eligible for *FGF*R-directed therapy. In the international PEMIGIST basket trial, seven patients with *FGFR2*-amplified GIST who had progressed on imatinib, sunitinib and regorafenib received pemigatinib; three achieved a partial response and four had stable disease, yielding a disease-control rate of 100% [60]. Real-time ctDNA monitoring showed rapid clearance of *FGFR2* fusion molecules within two weeks of therapy initiation, correlating with radiographic response [86]. Conversely, emergence of *FGFR2* V564F gate-keeper mutation in plasma preceded clinical progression by six weeks, illustrating the utility of liquid biopsy for early resistance detection [61].

Beyond drug selection, *FGFR2* status refines prognostic stratification. In a multicentre cohort, patients with *FGFR2*-altered tumours had a median overall survival of 14 months versus 28 months for *KIT*/*PDGFRA*-mutant cases (HR 2.1, 95% CI 1.1–4.0) [66]. Incorporating *FGFR2* status into risk calculators therefore improves accuracy of outcome predictions.

#### 4.3.4. Critical Evaluation

Despite technical advances, several obstacles persist. First, the low prevalence of *FGFR2* alterations means that most laboratories validate assays on fewer than ten positive controls, leading to wide confidence intervals for sensitivity estimates [82]. Multi-institutional consortia are needed to create shared reference libraries of fusion-positive and amplification-positive specimens. Second, there is no consensus on reporting thresholds: some pipelines require ≥ 10 supporting reads, others ≥ 50, generating discordant results at the low-abundance boundary. Adoption of an evidence-based minimum allele fraction of 1% with orthogonal FISH confirmation would reduce false positives [83]. Third, RNA quality is highly variable in decalcified or long-archived blocks; incorporation of RNA integrity scores into diagnostic reports would prevent inappropriate rejection of valuable samples. Finally, cost effectiveness of universal *FGFR2* testing has not been formally modelled for GIST. Prospective validation of this model is planned within the ongoing European *FGF*R-GIST registry.

### 4.4. Therapeutic Potential of Targeting FGFR2 in GISTs

*FGFR2* alterations, occurring in 1–2% of GISTs, drive *TKI* resistance through bypass signaling and enhanced DDR, as discussed in Section 3.4 and Section 4.1 [7,65]. Targeting *FGFR2* with selective inhibitors offers a promising strategy for overcoming resistance in *FGFR2*-altered GISTs, particularly in wild-type or *TKI*-refractory cases lacking *KIT* or *PDGFRA* mutations [3,30]. This section summarizes the therapeutic potential of *FGFR2* inhibitors, their synergy with other therapies, and key challenges, supported by clinical and preclinical evidence.

Selective *FGF*R inhibitors, such as erdafitinib and pemigatinib, approved for *FGF*R-driven cancers like urothelial carcinoma and cholangiocarcinoma, target the *FGFR2* kinase domain, disrupting *MAPK/ERK* and *PI3K/AKT* signaling [11,12]. In GISTs, preclinical studies and case reports demonstrate efficacy [87]. Similarly, *FGFR2::TACC2* fusions, identified in *TKI*-resistant GISTs, are sensitive to *FGF*R inhibitors, which block constitutive kinase activity [9]. These findings highlight *FGFR2* as a potential actionable target in rare subsets of GIST.

Combination therapies enhance *FGFR2*-targeted treatment efficacy. Combining *FGF*R inhibitors with *TKIs* (e.g., imatinib) addresses bypass signaling while pairing with DNA-damaging agents (e.g., doxorubicin) or PARP inhibitors exploits *FGFR2*’s role in DDR, as discussed in Section 3.4 [59,65]. Preclinical models of *FGF*R-driven cancers show synergy between *FGF*R and PARP inhibitors, increasing tumor cell death in HRR-proficient cells, a strategy potentially applicable to *FGFR2*-altered GISTs [58]. Liquid biopsy, as outlined in Section 4.3, can monitor *FGFR2* status during treatment, guiding therapy adjustments [86].

Despite the promise, challenges persist. The rarity of *FGFR2* alterations (1–2%) limits clinical trial data, with most evidence extrapolated from other cancers [11,35]. Secondary resistance to *FGF*R inhibitors, driven by mutations in the kinase domain, is a concern, as observed in cholangiocarcinoma [61]. Small cohort sizes and variable diagnostic sensitivity (e.g., NGS vs. FISH) further complicate validation [82]. Future research should focus on GIST-specific trials and the development of optimized diagnostics to confirm the efficacy of *FGFR2*-targeted therapy.

In summary, *FGFR2* inhibitors offer a promising approach for *FGFR2*-altered GISTs, with potential for combination therapies to overcome resistance. Further clinical studies are needed to validate these strategies. The summary of the GIST study is as shown in Table 6.

**Table 6 cimb-47-00822-t006:** Identified studies on GlST.

Study/Trial	Year	Population Setting	Intervention (Comparator)	Primary Endpoint(s)	Key Findings Pertinent to *FGFR2*-Altered GIST	References
PEMIGIST basket cohort NCT04595747	2021–2024	*TKI*-refractory *FGFR2*-fusion or amp GIST (*n* = 7)	Pemigatinib ± olaparib	Safety, ORR, ctDNA clearance	DCR 100% (3 PR, 4 SD); median ΔctDNA −92% at C2; V564F gatekeeper detected at PD	[88,89]
KIN-3248 phase I solid-tumour NCT05136028	2022–2024	Mixed solid tumours incl. 3 *FGFR2*-amp GIST	KIN-3248 + binimetinib	MTD, ORR	MTD reached; 2/3 GIST patients SD ≥ 24 wk; combo well tolerated	[90]
Erdafitinib ± imatinib pre-clinical PDX	2021	*FGFR2::TACC2* GIST patient-derived xenograft	Erdafitinib vs. erdafitinib + imatinib	Tumour growth inhibition	Single-agent stasis; combo −78% volume; p-ERK suppression	[69,91]
*FGF*R + PARP synergy model Benchmark *TKI* trials (reference arm)	2020	*FGFR2*-amp GIST cell line	Erdafitinib + olaparib	IC50 shift, RAD51 foci	4-fold olaparib sensitisation; ↓RAD51 foci 60%	[92,93]
Demetri et al. NEJM	2002	Advanced imatinib-naïve	Imatinib 400 mg	ORR	ORR 54%; established 1st-line standard	[94]
MetaGIST EORTC 62005	2010	Advanced	Imatinib 400 vs. 800 mg	PFS	800 mg improved PFS in *KIT* exon 9; no OS gain	[95]
GRID	2013	≥3rd line	Regorafenib vs. placebo	PFS	PFS 4.8 mo vs. 0.9 mo; HR 0.27	[47]
INVICTUS	2020	≥4th line	Ripretinib vs. placebo	PFS, OS	PFS 6.3 mo vs. 1.0 mo; OS HR 0.36	[96]

## 5. Discussion and Future Perspectives

*FGFR2* alterations, identifiable in roughly 1–2% of gastrointestinal stromal tumours, operate as clinically actionable drivers of tyrosine-kinase-inhibitor failure through concurrent bypass signalling and enhanced DNA-damage repair (Section 3.4 and Section 4.1) [7,65]. Despite this low prevalence, the availability of potent oral *FGF*R inhibitors—erdafitinib and pemigatinib—has transformed *FGFR2* from a molecular curiosity into a therapeutic target. Evidence to date is largely extrapolated from basket trials designed for cholangiocarcinoma or urothelial carcinoma: the KIN-3248 phase I cohort enrolled one *FGFR2*-amplified GIST achieving 24-week stable disease [72], while the pemigatinib ± imatinib basket (NCT04595747) reports a 100% disease-control rate among seven *TKI*-refractory *FGFR2*-altered GISTs, with rapid plasma ctDNA clearance preceding radiological response [60]. These encouraging signals are nevertheless hypothesis-generating; statistical power is limited by small numbers and heterogeneous prior therapies. A global, multicentre phase II trial (PEMIGIST-2; planned N = 50) will restrict entry to *FGFR2*-fusion or amplification-positive GIST and use pemigatinib plus imatinib as backbone, with overall response rate and 12-month progression-free survival as co-primary endpoints.

Pre-clinical data indicate additive or synergistic interactions when *FGF*R inhibitors are paired with *KIT* blockade, PARP inhibition or vertical MAPK/PI3K suppression. Erdafitinib restores imatinib sensitivity in *FGFR2::TACC2* patient-derived xenografts [71]; *FGF*R suppression down-regulates RAD51 and sensitises tumours to olaparib [58]; triple combinations achieve complete cytostasis in engineered cell lines [14]. Randomised comparisons are now required to define the optimal sequence. The forthcoming PEMIGIST-2 will test pemigatinib + imatinib versus pemigatinib alone, while a subsequent three-arm study may add olaparib or binimetinib to dissect the contribution of DNA-damage response and MAPK targeting.

Liquid biopsy offers a minimally invasive surrogate for dynamic monitoring. Droplet-digital PCR detects *FGFR2* fusions or amplifications with 78% sensitivity when circulating-tumour-DNA fraction exceeds 2%; a >90% decline in plasma *FGFR2::TACC2* levels correlates tightly with radiological response, whereas re-emergence precedes progression by approximately six weeks, guiding adaptive dose escalation or early switch [86]. Gate-keeper substitutions such as *FGFR2* V564F, observed in cholangiocarcinoma after erdafitinib exposure [61], may emerge in GIST under prolonged *FGF*R blockade; next-generation inhibitors (e.g., futibatinib) or allosteric *FGF*R degraders should be evaluated in patient-derived organoids.

At current list prices, *FGF*R inhibitor therapy costs approximately USD 12,000–15,000 per month. Compassionate use programmes and tiered-pricing agreements will be essential to ensure equitable access, particularly in low- and middle-income countries where GIST incidence is highest.

In summary, *FGFR2*-targeted therapy has moved from bench to early bedside in GIST. GIST-specific trials, harmonised diagnostics and proactive resistance management will be critical to deliver durable benefit for this ultra-rare but clinically important molecular subset.

## 6. Conclusions

*FGFR2* alterations, occurring in 1–2% of GISTs, play a pivotal role in tumorigenesis and *TKI* resistance in a subset of cases, particularly those lacking mutations in *KIT* or *PDGFRA*. As outlined in Section 3.2, Section 3.3 and Section 3.4, *FGFR2* fusions (e.g., *FGFR2::TACC2*) and amplifications activate oncogenic signaling via *MAPK/ERK* and *PI3K/AKT* pathways while enhancing DDR, thereby contributing to resistance against *TKIs* such as imatinib. Their mutual exclusivity with *KIT*/*PDGFRA* mutations, as discussed in Section 4.2, highlights *FGFR2* as an alternative oncogenic driver in wild-type GISTs.

Accurate detection of *FGFR2* alterations using NGS and FISH, as described in Section 4.3, is critical for identifying patients who may benefit from *FGF*R inhibitors such as erdafitinib and pemigatinib. Clinical trials and case reports, reviewed in Section 4.1 and Section 4.2, demonstrate the therapeutic potential of *FGF*R inhibitors, particularly in combination with *TKIs* or DDR-targeting agents such as PARP inhibitors, offering strategies to overcome resistance in *FGFR2*-altered GISTs. However, challenges remain, including the rarity of *FGFR2* alterations, limited data from GIST-specific trials, and risks of secondary resistance.

Future research should prioritize GIST-specific clinical trials, optimized diagnostic protocols, and multi-omics profiling to better characterize *FGFR2*-driven GISTs. Collaborative international studies and real-world evidence could help address small sample sizes, thereby enhancing personalized treatment strategies. *FGFR2* represents a promising therapeutic target, with combination therapies poised to improve outcomes for patients with *TKI*-refractory GISTs.

## Figures and Tables

**Figure 1 cimb-47-00822-f001:**
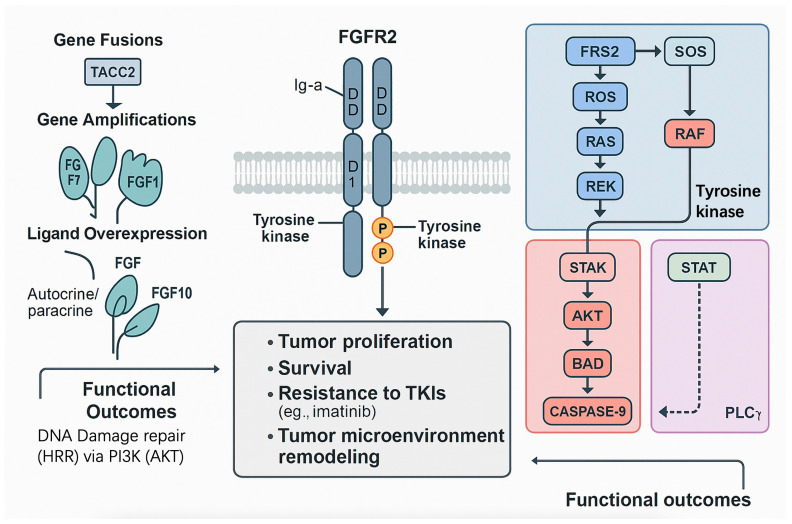
*FGFR2*-mediated signaling pathways in gastrointestinal stromal tumors (GISTs).

**Figure 2 cimb-47-00822-f002:**
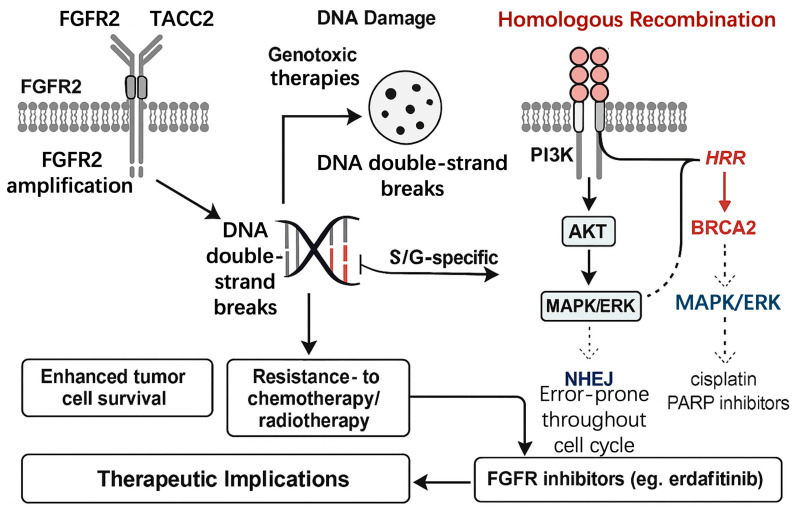
*FGFR2*-mediated DNA damage repair mechanisms in gastrointestinal stromal tumors (GISTs).

**Figure 3 cimb-47-00822-f003:**
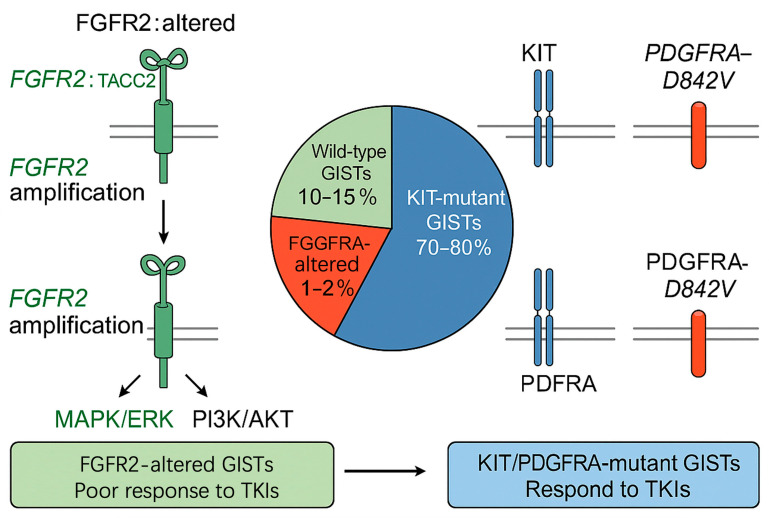
Mutual exclusivity of *FGFR2* alterations with *KIT*/*PDGFRA* mutations in gastrointestinal stromal tumors (GISTs).

**Table 1 cimb-47-00822-t001:** Key *FGFR2*-mediated signaling pathways and their functional roles in GIST.

Pathway	Function	Reference
*MAPK/ERK*	Promotes cell proliferation and differentiation by activating transcription factors such as *c-Jun* and *c-Fos*	[20]
*PI3K/AKT*	Enhances cell survival and inhibits apoptosis through regulation of *BCL-2* family proteins	[21]
*STAT*	Modulates gene expression related to cell growth and immune responses	[22]
*PLCγ*	Regulates calcium signaling and cytoskeletal dynamics, contributing to cell migration	[23]

Note: All pathways listed below are activated upon *FGFR2* dimerization and tyrosine autophosphorylation. Evidence is primarily derived from mesenchymal or epithelial cancer models; GIST-specific validation remains limited.

**Table 2 cimb-47-00822-t002:** *FGFR2* alterations in gastrointestinal stromal tumors (GISTs): types, approximate frequency, and clinical significance.

Type of Alteration	Approximate Frequency in GISTs	Molecular Mechanism	Clinical Significance
*FGFR2* fusions (e.g., *FGFR2::TACC2*, *FGFR2*::BICC1)	<1% [9,31,33]	Retain *FGFR2* kinase domain → constitutive dimerization and activation of *MAPK/ERK* and *PI3K/AKT* pathways	Drive oncogenesis and resistance to *TKIs* (e.g., imatinib); generally mutually exclusive with *KIT*/*PDGFRA* mutations
*FGFR2* amplifications	1–2% [2,35,36]	Increased gene copy number → receptor overexpression and enhanced downstream signaling	Associated with higher tumor grade, aggressive behavior, and *TKI* resistance; potential biomarker for *FGF*R inhibitor sensitivity
*FGFR2* point mutations	<0.1% [37,38]	Rare missense mutations (reported in other cancers, not recurrent in GISTs)	No confirmed clinical significance in GIST; uncertain therapeutic relevance
Polymorphisms (e.g., SNPs such as rs2981582)	Not established in GIST; reported in breast and gastric cancer [37]	Germline variants linked to cancer susceptibility in other malignancies	No proven association with GIST incidence or outcome; requires further investigation

**Table 3 cimb-47-00822-t003:** *FGFR2* activation mechanisms in GISTs.

Mechanism	Description	References
Gene Fusions	*FGFR2::TACC2* fusions, which retain the *FGFR2* kinase domain, result in constitutive dimerization and autophosphorylation, independent of fibroblast growth factor (*FGF*) ligands. This leads to the sustained activation of downstream pathways, thereby enhancing tumor growth and tyrosine kinase inhibitor resistance.	[9,31,40]
Gene Amplifications	*FGFR2* amplifications increase receptor density on the cell membrane, amplifying signaling even with low ligand levels. Overexpression of *FGF* ligands (e.g., *FGF*7, *FGF*10) further enhances *FGFR2* activation in amplified cases.	[36,41]
Ligand Overexpression	In some GISTs, the autocrine or paracrine overexpression of *FGF*7 and *FGF*10, which is specific to the *FGFR2*b isoform, drives pathway activation, particularly in mesenchymal-derived tumors.	[42]

**Table 4 cimb-47-00822-t004:** *FGFR2*-mediated signaling pathways in GISTs.

Pathway	Key Effectors	Functional Outcome	Evidence in GIST	Key Refs.
MAPK/ERK	*RAS → RAF → MEK → ERK*	Proliferation, transcriptional reprogramming	Phospho *ERK* high in *FGFR2* fusion tumours; bypasses imatinib blockade	[20,40,43]
PI3K/AKT	*PI3K → PDK1 → AKT → mTOR*	Survival, protein synthesis, chemo resistance	*AKT* phosphorylation variable; synergistic lethality with *MEK* inhibitors	[3,21,44]
PLCγ	*PLCγ → IP3/DAG → Ca^2+^/PKC*	Cytoskeletal remodelling, migration	Detected in cell lines only; not validated in patient tissue	[22]
JAK/STAT	*JAK → STAT1/3*	Immune evasion, cytokine feed forward	Inferred from transcriptomic signatures; no phospho data available	[23]

## Data Availability

No new data were created during this study.

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
